# MorphTE: Injecting Morphology in Tensorized Embeddings

**Guobing Gan**[1], **Peng Zhang**[1]*, **Sunzhu Li**[1], **Xiuqing Lu**[1], **and Benyou Wang**[2,3]

[1]College of Intelligence and Computing, Tianjin University, Tianjin, China
[2]School of Data Science, The Chinese University of Hong Kong, Shenzhen, China
[3]Shenzhen Research Institute of Big Data, The Chinese University of Hong Kong, Shenzhen, China
{ganguobing,pzhang,lisunzhu,lvxiuqing}@tju.edu.cn, wangbenyou@cuhk.edu.cn

## Abstract

In the era of deep learning, word embeddings are essential when dealing with text tasks. However, storing and accessing these embeddings requires a large amount of space. This is not conducive to the deployment of these models on resource-limited devices. Combining the powerful compression capability of tensor products, we propose a word embedding compression method with morphological augmentation, **Morphologically-enhanced Tensorized Embeddings** (**MorphTE**). A word consists of one or more **morphemes**, the smallest units that bear meaning or have a grammatical function. MorphTE represents a word embedding as an entangled form of its morpheme vectors via the **tensor product**, which injects prior semantic and grammatical knowledge into the learning of embeddings. Furthermore, the dimensionality of the morpheme vector and the number of morphemes are much smaller than those of words, which greatly reduces the parameters of the word embeddings. We conduct experiments on tasks such as machine translation and question answering. Experimental results on four translation datasets of different languages show that MorphTE can compress word embedding parameters by about 20 times without performance loss and significantly outperforms related embedding compression methods.

## 1 Introduction

The word embedding layer is a key component of the neural network models in natural language processing (NLP). It uses an embedding matrix to map each word into a dense real-valued vector. However, when the vocabulary size and word embedding size (dimensionality) are large, the word embedding matrix requires a large number of parameters. For example, the One Billion Word task of language modeling [8] has a vocabulary size ($|V|$) of around $800K$. Besides, the embedding size ($d$) can range from 300 to 1024 [32, 11, 23]. Storing and accessing the $|V| \times d$ embedding matrix requires a large amount of disk and memory space. This limits the deployment of these models on such devices having limited resources. To resolve this issue, there are many studies compressing embedding layers [34, 15, 29]. They can be roughly divided into two lines: **product quantization-based** and **decomposition-based** methods. The product quantization-based methods [34, 20, 38] mainly utilize the compositional coding for constructing the word embeddings with fewer parameters, and it needs to introduce an additional task to learn the compact code for each word.

The decomposition-based word embedding compression methods are mostly based on low-rank matrix factorization [18, 9] and tensor decomposition [15, 29]. Utilizing low-rank matrix factorization,

---

*Corresponding Author
[1]Code available at URL: `https://github.com/bigganbing/Fairseq_MorphTE`

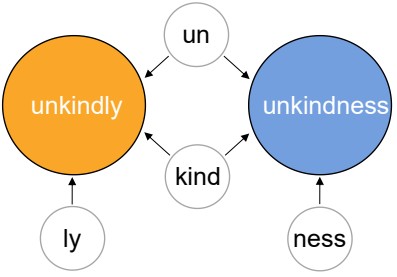

Figure 1: Morphemes of "unkindly" and "unkind-ness".

Table 1: Phenomena in word formation.

| Phenomenon | Example |
|---|---|
| **Inflection** | cook+s, cook+ed, cook+ing
cold+er, cold+est |
| **Derivation** | un+like, un+like+ly
im+poss+ible, im+poss+ibly |
| **Compounding** | police+man, post+man
cuttle+fish, gold+fish |

ALBERT [18] replaces the embedding matrix with the product of two small matrices. The tensor decomposition is widely used for parameter compression [15, 22, 25, 29, 41]. Inspired by quantum entanglement, Word2ket and Word2ketXS embeddings are proposed [29]. Specifically, Word2ket represents a word embedding as an entangled tensor of multiple small vectors (tensors) via the tensor product. The entangled form is essentially consistent with Canonical Polyadic decomposition [16]. Decomposition-based methods approximate the original large word embedding matrix with multiple small matrices and tensors. However, these small matrices or tensors have *no specific meaning* and *lack interpretation*, and the *approximate substitution* with them often hurts the model performance in complicated NLP tasks such as machine translation [15, 29].

In this study, we focus on high-quality compressed word embeddings. To this end, we propose the *Morphologically-enhanced Tensorized Embeddings* (**MorphTE**), which injects morphological knowledge in tensorized embeddings. Specifically, MorphTE models the embedding for a word as the entangled form of their **morpheme** vectors via **tensor products**. Notably, the quality of word embeddings can be improved by fine-grained morphemes, which has been verified in literature [4, 5].

The benefits of introducing the morphology of morphemes in MorphTE can be summed up in two points. **(1)** A word consists of morphemes which are considered to be the smallest meaning-bearing or grammatical units of a language [26]. As shown in Figure 1, the root 'kind' determines the *underlying meanings* of 'unkindly' and 'unkindness'. The affixes 'un', 'ly', and 'ness' *grammatically* refer to negations, adverbs, and nouns, respectively. In MorphTE, using these meaningful morphemes to generate word embeddings explicitly injects prior semantic and grammatical knowledge into the learning of word embeddings. **(2)** As shown in Table 1, linguistic phenomena such as inflection and derivation in word formation make morphologically similar words often semantically related. In MorphTE, these similar words can be connected by sharing the same morpheme vector.

MorphTE only needs to train and store morpheme vectors, which are smaller in embedding size and vocabulary size than original word embeddings, leading to fewer parameters. For example, a word embedding of size $512$ can be generated using three morpheme vectors of size $8$ via tensor products. In addition, since morphemes are the basic units of words, the size of the morpheme vocabulary is smaller than the size of the word vocabulary. To sum up, MorphTE can learn high-quality and space-efficient word embeddings, combining the prior knowledge of morphology and the compression ability of tensor products.

We conducted comparative experiments on machine translation, retrieval-based question answering, and natural language inference tasks. Our proposed MorphTE achieves better model performance on these tasks compared to related word embedding compression methods. Compared with Word2ket, MorphTE achieves improvements of $0.7$, $0.6$, and $0.6$ BLEU scores on De-En, En-It, and En-Ru datasets respectively. In addition, on $4$ translation datasets in different languages, our method can maintain the original performance when compressing the number of parameters of word embeddings by more than $20$ times and reducing the proportion of word embeddings to the total parameters approximately from $30\%$ to $2\%$, while other compression methods hurt the performance.

The main contributions of our work can be summarized as follows:

- We propose MorphTE, a novel compression method for word embeddings using the form of entangled tensors with morphology. The combination of morpheme and tensor product can compress word embeddings in terms of both vocabulary and embedding size.

- MorphTE introduces prior semantic knowledge in the learning of word embeddings from a fine-grained morpheme perspective, and explicitly models the connections between words by sharing morpheme vectors. These enabled it to learn high-quality compressed embeddings.
- Experiments on multiple languages and tasks show that MorphTE can compress word embedding parameters over 20 times without hurting the original performance.

## 2 Related Work

**Morphologically-augmented Embeddings.** Related works [24, 4, 5, 31, 2, 10] propose to improve the quality of word embeddings by integrating morphological information. Representing word embeddings as the sum of morpheme and surface form vectors has been employed in several studies [5, 31, 2]. Morphological RNNs [5] learns word representations using morphemes as units of recursive neural networks [35]. Our proposed MorphTE also utilizes the information of morphemes and is a decomposition-based word embedding compression method, similar to Word2ket [29].

**Decomposition-based Compression Embeddings.** Decomposition-based methods are either based on low-rank matrix factorization [18, 9, 1, 21, 19] or tensor decomposition [15, 44, 29, 36]. Based on low-rank matrix factorization, ALBERT [18] simply approximates the embedding matrix by the product of two small matrices. GroupReduce [9] and DiscBlock [19] perform a fine-grained matrix factorization. They first block the word embedding matrix according to the word frequency and then approximate each block. Notably, the method based on matrix factorization has a low-rank bottleneck, and its expressive ability is limited under the condition of a high compression ratio [37].

As for the tensor decomposition, TT embeddings [15] uses the Tensor Train decomposition [27] to approximate the embedding matrix with several 2-order and 3-order tensors. TT-Rec [44] improves TT embeddings in terms of implementation and initialization to fit the recommended scenario. Word2ket [29] represents a word embedding as an entangled tensor via multiple small vectors. It essentially exploits Canonical Polyadic decomposition [16, 17]. Word2ketXs [29] is similar to Word2ket, but it compresses embeddings from the perspective of all words rather than individual words. In addition, KroneckerBERT [36] uses Kronecker decomposition to compress the word embeddings, and the form of Kronecker Embeddings is consistent with Word2ket [29] with an order of 2. Unlike these compression methods, our MorphTE utilizes meaningful morphemes as basic units for generating word embeddings, rather than vectors or tensors with no specific meaning.

## 3 Preliminary

### 3.1 Tensor Product Space and Entangled Tensors

A tensor product space of two separable Hilbert spaces $\mathcal{V}$ and $\mathcal{W}$ is also a separable Hilbert space $\mathcal{H}$, which is denoted as $\mathcal{H} = \mathcal{V} \otimes \mathcal{W}$. Suppose $\{\psi_1, \ldots, \psi_g\}$ and $\{\phi_1, \ldots, \phi_h\}$ are the orthonormal basis in $\mathcal{V}$ and $\mathcal{W}$, respectively. The tensor product of the vector $c = \sum_{j=1}^{g} c_j \psi_j \in \mathcal{V}$ and $e = \sum_{k=1}^{h} e_k \phi_k \in \mathcal{W}$ is defined as follow:

$$c \otimes e = \left\{ \sum_{j=1}^{g} c_j \psi_j \right\} \otimes \left\{ \sum_{k=1}^{h} e_k \phi_k \right\} = \sum_{j=1}^{g} \sum_{k=1}^{h} c_j e_k \psi_j \otimes \phi_k \tag{1}$$

The set $\{\psi_j \otimes \phi_k\}_{jk}$ forms the orthonormal basis in $\mathcal{H}$, and the dimensionality of $\mathcal{H}$ is the product $(gh)$ of dimensionalities of $\mathcal{V}$ and $\mathcal{W}$. This tensor product operation can be simplified as the product of the corresponding coefficients as follow:

$$\begin{aligned} c \otimes e &= [c_1, c_2, \ldots, c_g] \otimes [e_1, e_2, \ldots, e_h] \\ &= [\underline{c_1 e_1, c_1 e_2, \ldots, c_1 e_h}, \ldots, \underline{c_g e_1, c_g e_2, \ldots, c_g e_h}] \end{aligned} \tag{2}$$

The cumulative tensor product space of the following form is said to have a tensor **order** of $n$, and the dimensionality of cumulative tensor product space is the cumulative product of its subspace dimensionalities. See Appendix B for concrete examples of the cumulative tensor product of multiple vectors.

$$\bigotimes_{j=1}^{n} \mathcal{H} = \mathcal{H}_1 \otimes \mathcal{H}_2 \ldots \otimes \mathcal{H}_n \tag{3}$$

Considering the $n$-order tensor product space $\bigotimes_{j=1}^{n} \mathcal{H}_j$, vectors of the form $v = \otimes_{j=1}^{n} v_j$, where $v_j \in \mathcal{H}_j$, are called **simple tensors**. In addition, vectors need to be represented as the sum of multiple simple tensors are called **entangled tensors**. Tensor **rank** of a vector $v$ is the smallest number of simple tensors that sum up to $v$.

## 3.2 Tensorized Embeddings with Tensor Product

Tensor products have been introduced to learn parameter-efficient word embeddings in KroneckerBERT [36] and Word2ket[29]. As shown in Figure 2, Word2ket[29] represents the embedding $v \in \mathbb{R}^d$ of a word as an entangled tensor of rank $r$ and order $n$ as follow:

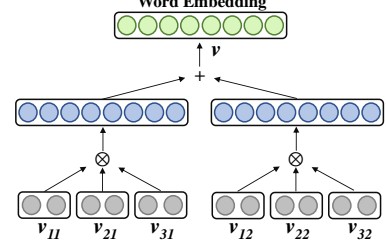

**Word Embedding**

Figure 2: Word2ket embedding with a rank of $r = 2$ and an order of $n = 3$.

$$v = \sum_{k=1}^{r} \bigotimes_{j=1}^{n} v_{jk} \qquad (4)$$

where $v_{jk} \in \mathbb{R}^q$ and $v \in \mathbb{R}^{q^n}$. Word2ket only needs to store and use these small vectors $v_{jk}$ to generate a large word embedding. If $q^n > d$, the excess part of the generated embedding will be cut off. Therefore, setting $q^n = d$ can avoid the waste of parameters caused by clipping, and the number of embedding parameters for a word is reduced from $d$ to $rn\sqrt[n]{d}$. For example, when $d = 512$, $q = 8$, $n = 3$, and $r = 2$, the number of parameters of a word embedding can be reduced from $512$ to $48$.

# 4 Methodology: Morphologically-enhanced Tensorized Embeddings

In this section, we first discuss the rationale for introducing morphology in the embedding compression. Then, we propose **MorphTE**, a morphologically-enhanced word embedding compression method based on the tensor product. Finally, we show the detailed workflow of MorphTE.

## 4.1 Motivation to Introduce Morphology

To achieve compression, existing decomposition-based word embedding compression methods [18, 15] use a series of small vectors or tensors to generate large word embeddings, as shown in Figure 2. These methods are not only uninterpretable as their small tensors do not have specific meaning [15, 29], but also lack lexical knowledge. We argue that, in resource-limited scenarios like compression, knowledge injection is much more critical than in common scenarios. Since with a significant amount of parameters in common scenarios it could to an easier extent learn implicitly such knowledge in a data-driven way, which is also one of the objectives for neural networks. However, in compression, it is more beneficial to inject explicit knowledge to compensate for inferiority in parameter scales, therefore underscoring *the importance of knowledge injection in compression*.

From a *reductionism* point of view, words might not be the smallest unit for some languages; for example `unfeelingly` could be separated into four meaningful parts [`un`, `feel`, `ing`, `ly`], a.k.a., morphemes [2]. By using a limited number of morphemes, one could possibly exponentially extend a given core vocabulary by composing morphemes as new words according to the rules of word formation in Table 1. The adoption of morphemes largely reduces the memory burden and therefore facilitates the learning of words for humans. We hypothesize that morphology also helps for word representation in neural networks, especially in resource-limited scenarios like compression.

## 4.2 Definition of MorphTE

Considering the above analysis, we propose to inject morphological knowledge to achieve high-quality and space-efficient word embeddings. Suppose a word is segmented as $l$ morphemes $[m_1, m_2, \ldots, m_l]$ in the natural order. For example, a four-morpheme word `unfeelingly` is

---

[2]This also holds for *logogram*, written symbols of which represent words instead of sounds. For example, Chinese language has character components, a.k.a, radicals.

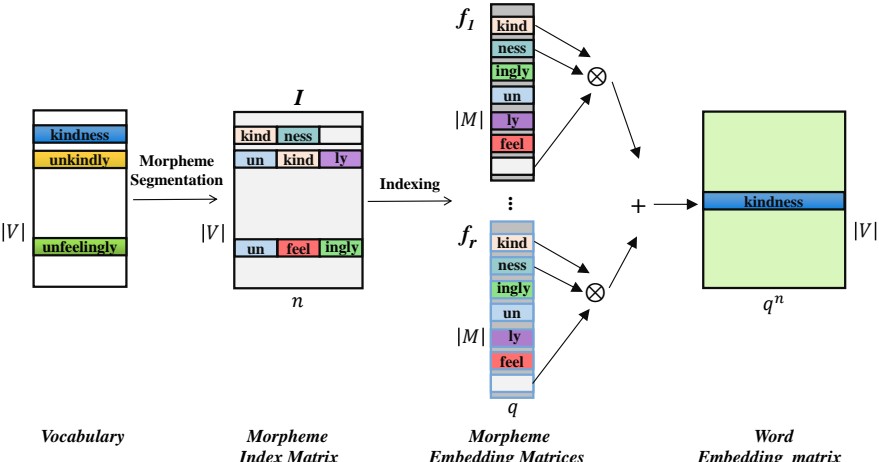

*Vocabulary*     *Morpheme Index Matrix*     *Morpheme Embedding Matrices*     *Word Embedding matrix*

Figure 3: The workflow of MorphTE. $n$ is the order (number of morphemes for a word). $q$ is the size of morpheme vectors. $|V|$ and $|M|$ denote the size of word vocabulary and morpheme vocabulary.

segmented as $[\mathtt{un}, \mathtt{feel}, \mathtt{ing}, \mathtt{ly}]$. We refer $f_1(\cdot), f_2(\cdot), \cdots f_r(\cdot) : \mathbb{N} \to \mathbb{R}^q$ as $r$ different *representation functions* of morphemes [3], selected from a parametric family $\mathcal{F} = \{f : \mathbb{N} \to \mathbb{R}^q\}$. The Morphologically-enhanced Tensorized Embedding (MorphTE in short) of a word is defined as a sum of cumulative tensor products of morpheme vectors from different representation functions, namely,

$$v = \sum_{i=1}^{r} \bigotimes_{j=1}^{l} f_i(m_j) \tag{5}$$

Each representation function of morphemes $f_i$ could sometimes be considered as a subspace of morphemes. The number of subspaces (i.e., $r$) is called the **rank**, and the number of morphemes (i.e., $n$) of a word is called the **order**, similar to Word2ket. The sum of the outputs from different subspaces in MorphTE is similar to the multi-head mechanism in Transformer [39].

**Reasons to use tensor product.** MorphTE utilizes tensor products to aggregate several small vectors Interestingly, we find that there are some commonalities between tensor product and morpheme composition. **(1)** Both tensor product and morpheme composition are non-commutative (e.g., $c \otimes e \neq e \otimes c$ and likewise "houseboat" $\neq$ "boathouse"). **(2)** Small vectors are the smallest units in Word2ket and morphemes are also the smallest meaning-bearing units of words. Naturally, these small vectors in Word2ket can be assigned morpheme identities and shared among different words.

### 4.3 Workflow of MorphTE

We describe the workflow of MorphTE in Figure 3.

**Morpheme segmentation** Suppose there are $|V|$ individual words. We first segment each word as a sequence of morphemes (a $l$-morpheme word could be segmented as a sequence as $[m_1, \ldots, m_l]$) using polyglot [4] To facilitate processing in neural networks, we truncate morpheme sequences in a fixed length $n$ (e.g., $n = 3$ or $n = 4$). **(1)** For those words that have less than $n$ morphemes, we pad them with some extra tokens; for example, we pad a single-morpheme word with $n-1$ padding morphemes. **(2)** For those words that have more than $n$ morphemes, we concatenate the rest of

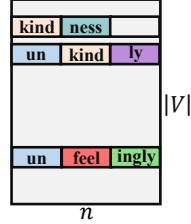

Figure 4: Segmentation

---

[3] For example, such a function could be morpheme embeddings, each vector of which is a $q$-sized vector, which are similar to word embeddings.

[4] polyglot could be found in `https://github.com/aboSamoor/polyglot`. Based on morfessor [40], polyglot provides trained morpheme segmentation models for a variety of languages. Numerous studies [3, 31, 12] also utilize morfessor for morpheme segmentation.

morphemes as the last one. The processing could be explained as follows:

$$
\begin{cases}
[m_1, m_2, \ldots, m_l, \mathrm{pad}_{l+1}, \ldots \mathrm{pad}_n], & l < n \\
[m_1, m_2, \ldots, \mathrm{concat}(m_n, \ldots, m_l)], & l > n
\end{cases}
\tag{6}
$$

where $\mathrm{concat}(\cdot)$ is the operation to concatenate multiple morphemes into one morpheme. For example, a four-morpheme word $\mathtt{unfeelingly}$, segmented as $[\mathtt{un}, \mathtt{feel}, \mathbf{ing}, \mathbf{ly}]$, could be then truncated into $[\mathtt{un}, \mathtt{feel}, \mathbf{ingly}]$ when $n = 3$.

After morpheme segmentation, we could get a $n$-length morpheme sequence for each word; this results in a matrix $I \in \mathbb{R}^{|V| \times n}$, each row of which (denoted as $I_j \in \mathbb{R}^n$ ) is the morpheme sequence for a word (i.e., $w_j$), see $I$ in Figure 4.

**Rank-one MorphTE** Assume that there are $|M|$ individual morphemes. We first define a $q$-sized trainable morpheme embeddings $f \in \mathbb{R}^{|M| \times q}$. For a word $w_j$ with a morpheme sequence $[I_{j,1}, I_{j,2}, \cdots, I_{j,n}]$, its rank-one MorphTE embedding is a tensor product between these $q$-sized embeddings of these morphemes, resulting in a $q^n$-sized vector. Namely, the formula for a rank-one version of MorphTE is as follows:

$$
\mathrm{Embed}(w_j) = f(I_{j,1}) \otimes f(I_{j,2}) \otimes \ldots \otimes f(I_{j,n})
\tag{7}
$$

If $q^n$ is greater than the given dimensionality of word embeddings (i.e., $d$), the excess part of the generated word embedding will be discarded.

**General MorphTE** Here, we propose a general version of MorphTE; the rank-one MorphTE is a special case of MorphTE when its rank $r = 1$. We define $r$ copies of morpheme embeddings, namely, $f_1, \ldots, f_r \in \mathbb{R}^{|M| \times q}$ with Xavier [13]. The general MorphTE could be considered as a sum of $r$ rank-one MorphTE embeddings, $r$ is called a 'rank' of MorphTE. Technically, it is formalized as:

$$
\mathrm{Embed}(w_j) = \sum_{i=1}^{r} f_i(I_{j,1}) \otimes \ldots \otimes f_i(I_{j,n})
\tag{8}
$$

## 4.4 Difference between MorphTE and Word2ket

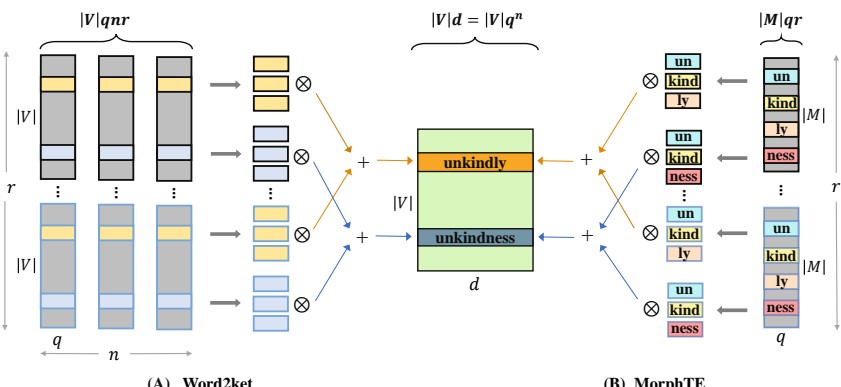

(A) Word2ket

(B) MorphTE

Figure 5: The graphical representation and parameters of Word2ket (A), and MorphTE (B). We show the procedure of how Word2ket and MorphTE generate word embeddings. *|V|* and *|M|* denote the size of word vocabulary and morpheme vocabulary. *d* indicates the word embedding size and *q* indicates the size of the small vector in Word2ket and MorphTE. *r* is the rank and *n* is the order.

Although Eq. 5 of MorphTE and Eq. 4 of Word2ket have a similar form, these two methods are fundamentally different. Compared with Word2ket, the innovations of MorphTE are as follows:

**MorphTE is morphologically-enhanced.** MorphTE uses morpheme vectors with meaning or grammatical functions to generate word embeddings. Compared with vectors without specific meanings in Word2ket, this way introduces prior morphological knowledge and has better interpretability.

**MorphTE captures commonness among words.** MorphTE can explicitly add connections of morphologically similar words by sharing morpheme vectors. As shown in Figure 5, in MorphTE,

"unkindly" and "unkindness" share the vectors of "un" and "kind", and they are both semantically related to "unkind". However, Word2ket considers these two words to be completely unrelated.

**MorphTE is more parameter-efficient.** Suppose MorphTE and Word2ket have the same rank ($r \geq 1$), order ($n \geq 2$), and dimensionality ($q$) of the subspace. **(1)** As shown in Figure 5, the number of trainable parameters required by MorphTE and Word2ket are $|M|qr$ and $|V|qnr$, respectively. **(2)** The number of parameters of Word2ket divided by that of MorphTE equals $n|V|/|M|$. **(3)** As smaller units, the number of morphemes ($|M|$) is usually smaller than the number of words ($|V|$). Referring to Table 5, $|V|/|M| > 2.5$. **(4)** Hence, MorphTE can save more than 5 times the parameters compared to Word2ket ($n|V|/|M| > 2.5n \geq 5$).

## 5 Experiments

### 5.1 Experimental Setup

**Baselines.** **(1)** For word embedding compression, we chose **Matrix Factor.** (low–rank matrix factorization [18]), **Tensor Train** [15], **Word2ket**, and **Word2ketXs** embeddings [29] for comparison. **(2)** For morphological word embeddings, we reproduced the method (called as **MorphSum**) of representing a word embedding as the sum of its morpheme and surface form vectors [5, 31, 2]. In addition, we reproduced the method (called as **MorphLSTM**) of using the output of the word's last morpheme in the LSTM network [14] as the embedding of the word, referring to Morphological RNNs [5]. Our MorphTE utilizes the same morpheme segmentation as MorphSum and MorphLSTM. Besides, we call the original word embeddings as **Original**. Except for the embedding layer, all models have the same structure.

**Tasks, Datasets, and Metrics.** We conducted experiments on machine translation, question answering, and natural language inference (NLI) tasks. **(1)** For machine translation tasks, we chose IWSLT'14 German-to-English (De-En) dataset [7], English-to-Italian (En-It), English-to-Spanish (En-Es), and English-to-Russian (En-Ru) datasets of OPUS-100 [45]. The De-En dataset contains 160K sentence pairs and is processed by the BPE [33] of 10K tokens. The En-It, En-Es, and En-Ru datasets contain 1M sentence pairs and use joint source-target vocabulary processed by the BPE of 40K tokens. The performance is measured by case-sensitive tokenized *BLEU*[30] for all translation tasks. **(2)** For question answering tasks, we chose *WikiQA* [43], a retrieval-based question answering dataset. It contains 20.4K training pairs, and *mean average precision* (*MAP*), *mean reciprocal rank* (*MRR*) are used for evaluation. **(3)** For NLI tasks, we chose *SNLI* [6] which consists of 570k annotated sentence pairs from an image captioning corpus. *Accuracy* is used as the evaluation metric.

**Implementations.** **(1)** For machine translation tasks, we chose Transformer [39] with the implementation of Fairseq [28]. For De-En dataset, the Transformer consists of 6-layer encoder and 6-layer decoder with 512 embedding size, 1024 feed-forward network (FFN) size. It is trained with a batch size of 4096 tokens on a NVIDIA Tesla V100 GPU. For En-It, En-Es, and En-Ru tasks, the FFN size is increased to 2048. They are trained with a batch size of 32768 tokens on 2 NVIDIA Tesla V100 GPUs. **(2)** For question answering and NLI tasks, we followed the implementation and setup of RE2 [42]. The word embedding size is set to 512, and we trained them for 30 epochs with the early stopping. **(3)** Notices: Unless otherwise specified, the hyperparameter *order* of MorphTE is 3 in our experiments. For MorphTE and word embedding compression baselines, they are compared under a roughly equal number of parameters (compression ratio) by adjusting their hyperparameters of the *rank*. For more details on hyperparameters and training settings, refer to Appendix D.

### 5.2 Main Results

**MorphTE outperforms compression baselines.** As shown in Table 2, we compared MorphTE and compression baselines at about $20\times$ and $40\times$ compression ratios on four translation datasets. **(1)** At the ratio of $20\times$, MorphTE can maintain the performance of the original embeddings on all datasets. Specifically, MorphTE achieves $0.4$ and $0.3$ BLEU score improvements on the De-En and En-Ru datasets, respectively. For other datasets, it achieves the same BLEU scores. However, none of the other compression methods can maintain the performance of the original model on these datasets. **(2)** At the ratio of $40\times$, although almost all compression methods cannot maintain the original performance, MorphTE still achieves the best results compared to other compression baselines.

Table 2: Experimental results of different embedding compression methods on translation tasks. **BLEU scores** and **compression ratios** of the embedding layers are reported, with the form of B(C×). - indicates that the method can not achieve approximately the same compression ratio as other methods. Refer to Appendix C for the analysis of the parameters of the methods.

| Method | De-En | | En-It | | En-Es | | En-Ru | |
|---|---|---|---|---|---|---|---|---|
| Original | 34.5 (1.0×) | - | **32.9** (1.0×) | - | **39.1** (1.0×) | - | 31.6 (1.0×) | - |
| Matrix Factor. | 32.7 (19×) | 22.8 (40×) | 31.0 (20×) | 23.2 (42×) | 38.0 (20×) | 29.7 (42×) | 28.9 (20×) | 17.9 (40×) |
| Tensor Train | 34.3 (20×) | 33.4 (43×) | 32.4 (21×) | **32.1** (43×) | 38.7 (21×) | 38.5 (43×) | 31.2 (21×) | 30.9 (41×) |
| Word2ketXs | 34.3 (21×) | 33.7 (42×) | 32.6 (21×) | 31.5 (43×) | 38.4 (21×) | 38.0 (43×) | 31.5 (21×) | 31.0 (41×) |
| Word2ket | 34.2 (21×) | - | 32.3 (21×) | - | **39.1** (21×) | - | 31.3 (21×) | - |
| MorphTE | **34.9 (21×)** | **34.1** (43×) | **32.9** (21×) | **32.1** (45×) | **39.1** (21×) | **38.7** (43×) | **31.9** (21×) | **31.5** (41×) |

Notably, the performance of the Matrix Factor. degrades significantly when a higher compression is conducted, and Word2ket cannot achieve more than 40× compression on these datasets.

**MorphTE can handle different languages and tasks well. (1)** We validated MorphTE on translation datasets of De-En, En-It, En-Es, and En-Ru. MorphTE has been shown to handle different languages effectively. **(2)** For translation tasks, the De-En dataset uses separate source and target dictionaries, while other datasets use a shared source-target dictionary. MorphTE is shown to work in both ways. **(3)** Besides translation tasks, we also validated MorphTE on question answering (WikiQA) and natural language inference (SNLI) tasks. The experimental results in Table 3 show that MorphTE still maintains the performance of the original model and outperforms other compression baselines on these different tasks.

Table 3: Experimental results on WikiQA of question answering tasks and SNLI of natural language inference tasks. The ratio means the compression ratio of embedding layers.

| Method | WikiQA | | | SNLI | |
|---|---|---|---|---|---|
| | ratio | MAP | MRR | ratio | Accuracy |
| Original | 1× | 0.6798 | 0.6970 | 1× | 0.8492 |
| Matrix Factor. | 82× | 0.5957 | 0.6121 | 38× | 0.4166 |
| Tensor Train | 80× | 0.6251 | 0.6440 | 37× | 0.8473 |
| Word2ketXs | 80× | 0.6686 | 0.6871 | 38× | 0.8450 |
| Word2ket | 21× | **0.6842** | 0.7025 | 21× | 0.8487 |
| MorphTE | 81× | 0.6834 | **0.7051** | 38× | **0.8497** |

Table 4: Experimental results for analyzing the effect of morphology. **BLEU scores** and **compression ratios** of the embedding layers are reported, with the form of B(C×). **Morph.** indicates whether the method introduces morphological knowledge. **Compr.** indicates whether the method supports high compression. **Word2ket+Rshare** means the method of random sharing of small vectors in Word2ket.

| Method | Morph. | Compr. | De-En | En-It | En-Es | En-Ru |
|---|---|---|---|---|---|---|
| Original | ✗ | ✗ | 34.5 (1.0×) | **32.9** (1.0×) | 39.1 (1.0×) | 31.6 (1.0×) |
| MorphSum | ✓ | ✗ | **34.9** (0.7×) | **33.0** (0.8×) | 39.1(0.8×) | 31.7 (0.8×) |
| MorphLSTM | ✓ | ✗ | **34.9** (1.6×) | 32.9 (2.8×) | **39.7** (2.7×) | **32.4** (2.5×) |
| Word2ket | ✗ | ✓ | 34.2 (21×) | 32.3 (21×) | 39.1 (21×) | 31.3 (21×) |
| Word2ket+Rshare | ✗ | ✓ | 34.0 (21×) | 32.0 (21×) | 38.3 (21×) | 30.6 (21×) |
| MorphTE | ✓ | ✓ | **34.9** (21×) | **32.9** (21×) | 39.1 (21×) | 31.9 (21×) |

**Morpheme-based morphology can enhance word embeddings.** To study the impact of morphemes on MorphTE, we conducted following explorations. **(1)** As introduced in Section 5.1, MorphSum and MorphLSTM both introduce morphological knowledge based on morphemes. As shown in Table 4, although they cannot achieve high compression on word embeddings, they (especially MorphLSTM) achieve significant improvements compared with original embeddings. This shows that morphological knowledge based on morphemes is beneficial for word embeddings. **(2)** MorphTE assigns the identities of morphemes to the small vectors of Word2ket, and shares these small vectors based on morphemes. We consider the method (called Word2ket+Rshare) of random sharing of small vectors in Word2ket rather than morpheme-based sharing. This method can be implemented in such a way that for each word, randomly assign several row vectors of the trainable matrix of the same shape as the morpheme embedding matrix. As shown in Table 4, Word2ket+Rshare has lower BLEU scores than Word2ket on different translation datasets, and it is significantly inferior to MorphTE. This also verifies that it makes sense to introduce morphemes in MorphTE.

Table 5: Statistics of translation tasks. $|V|$ and $|M|$ are the size of the word vocabulary and morpheme vocabulary, respectively. #Struc and #Emb are the number of parameters of the model structure and that of the embedding layer, respectively. ratio is the compression ratio of the word embedding layer. P is the proportion of the embedding layer to the total parameters.

| Dataset | $|V|$ | $|M|$ | #Stuc | Original | | MorphTE | | | | | | | |
|---|---|---|---|---|---|---|---|---|---|---|---|---|---|
| | | | | #Emb | P | ratio | #Emb | P | $\Delta$ BLEU | ratio | #Emb | P | $\Delta$ BLEU |
| **De-En** | 15480 | 5757 | 31.54M | 7.93M | 20.1% | 21× | 0.37M | 1.2% | + 0.4 | 43× | 0.18M | 0.6% | − 0.4 |
| **En-It** | 41280 | 10818 | 44.14M | 21.14M | 32.4% | 21× | 0.99M | 2.2% | + 0.0 | 45× | 0.45M | 1.0% | − 0.8 |
| **En-Es** | 41336 | 11377 | 44.14M | 21.16M | 32.4% | 21× | 1.03M | 2.3% | + 0.0 | 43× | 0.49M | 1.1% | − 0.4 |
| **En-Ru** | 42000 | 12423 | 44.14M | 21.50M | 32.8% | 21× | 1.02M | 2.3% | + 0.3 | 41× | 0.52M | 1.2% | − 0.1 |

Table 6: Experimental results for the ablation on the order of MorphTE on De-En. **d / q** means the size of the word embedding or morpheme vector. **|V| / |M|** means the size of the word or morpheme vocabulary. #Emb and ratio are the parameter number and compression ratio of the embedding layer, respectively.

| Method | order | rank | d / q | |V| / |M| | #Emb | ratio | BLEU |
|---|---|---|---|---|---|---|---|
| **Orginal** | - | - | 512 | 15480 | 7.93M | 1x | 34.5 |
| **MorphTE** | 2 | 2 | 23 | 7654 | 0.38M | 20x | 34.4 |
| | 3 | 7 | 8 | 5757 | 0.37M | 21x | **34.9** |
| | 4 | 11 | 6 | 5257 | 0.41M | 19x | 34.4 |

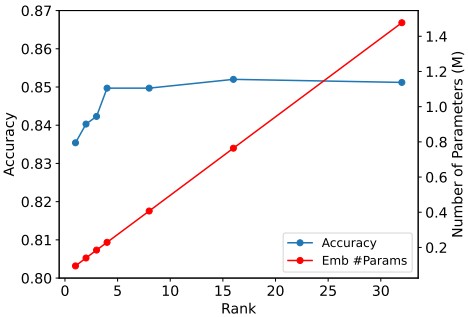

Figure 6: Accuracy on SNLI and parameter numbers of embeddings changes w.r.t. the rank.

**Marginal utility of higher compression ratios (e.g., $> 40$).** We report the statistics of translation tasks in Table 5. The experimental results of this table are consistent with those in Table 2. **(1)** At about $20\times$ compression, MorphTE can maintain the performance of the original word embeddings. The number of embedding parameters is approximately reduced from 20M to 1M, and its proportion of total parameters is reduced from 20.1%-31.8% to 1.2%-2.3%. **(2)** At about $40\times$ compression, although MorphTE outperforms other compression methods referring to Table 2, MorphTE's BLEU scores are significantly lower than the original model. Although the compression ratio is increased from $20\times$ to $40\times$, the parameters of embeddings are only approximately reduced from 1M to 0.5M, and the proportion of embedding parameters to the total parameters is only reduced from 1.2%-2.3% to 0.6%-1.2%. Considering the loss of model performance, the slight reduction in the number of parameters resulting from higher compression ratios does not make much sense.

## 5.3 Ablation Study

**Sensitivity on the *order*.** **(1)** In previous experiments, we set the order of MorphTE to 3. In this section, we perform the statistical analysis and ablation experiments on the De-En dataset. Morpheme statistics show that more than 90% of the words have no more than three morphemes, referring to Appendix E. **(2)** We set the order of MorphTE to 2, 3, and 4, that is, limiting the maximum number of morphemes to 2, 3, and 4, respectively. The word embedding size was set to 512. We adjusted the rank of MorphTE so that the models corresponding to these three cases have a similar number of parameters. From Table 6, setting the order of MorphTE to 3 achieves the best results when compressing the word embedding parameters by about 20 times.

**Sensitivity on the *rank*.** We show the effect of *rank* on the number of parameters and performance of MorphTE in Figure 6. The *rank* of MorphTE is set from 1 to 32. As can be seen, as the *rank* increases, the number of embedding parameters increases linearly, and the Accuracy first increases and then gradually stabilizes. This means that with the increase of embedding parameters, the model performance will gradually improve, but after the parameter amount exceeds a certain threshold, it will not drive the performance improvement. Therefore, the degree of compression and model performance can be balanced by adjusting the *rank* of MorphTE.

**Sensitivity on the *Embedding Size*.** To verify that MorphTE can stably achieve compression under different word embedding dimensionalities, We conducted experiments with the dimensionalities

Table 7: Experimental results for the ablation on the embedding size on IWSLT'14 De-En. $d/q$ means the size of the word embedding or morpheme vector. $|V|/|M|$ means the size of the word or morpheme vocabulary. $n$ and $r$ are the order and rank of MorphTE. **#Emb** and **#Stuc** are the parameter numbers of the embedding layer and the model structure, respectively. **ratio** is the compression ratio of word embedding layers.

| Method | $d/q$ | $n$ | $r$ | $|V|/|M|$ | #Emb | ratio | #Stuc | BLEU |
|--------|-------|-----|-----|-----------|------|-------|-------|------|
| **Original** | 216 | 3 | - | 15480 | 3.34M | 1× | 8.71M | 34.1 |
| **MorphTE** | 6 | 3 | 8 | 5757 | 0.32M | 10× | 8.71M | **34.2** |
| **Original** | 512 | 3 | - | 15480 | 7.93M | 1× | 31.54M | 34.5 |
| **MorphTE** | 8 | 3 | 8 | 5757 | 0.41M | 19× | 31.54M | **34.9** |
| **Original** | 1000 | 3 | - | 15480 | 15.48M | 1× | 96.72M | 32.8 |
| **MorphTE** | 10 | 3 | 8 | 5757 | 0.51M | 30× | 96.72M | **33.4** |

of 216, 512, and 1000. As can be seen in Table 7, when order $n = 3$ and the word embedding size / hidden size ($d$) is set to 216, 512, and 1000, the dimensionality of the morpheme vector ($q$) is set to 6, 8, and 10, respectively. In addition, the rank is set to 8 for these three cases. It can be seen that as the dimensionality of a word embedding increases significantly, the dimensionality of the morpheme vector does not increase significantly. This means that the larger the word embedding size, the more the compression potential of our method can be released. Specifically, when the word embedding dimensionality is 216, our method can compress the word embedding parameters by a factor of 10 with comparable model performance. When the word embedding dimensionality is set to 512, our method is able to compress the word embedding parameters by a factor of 19, with a 0.4 BLEU score higher than the uncompressed model. When the word embedding dimensionality is set to 1000, our method outperforms the original model by a 0.6 BLEU score when compressing the word embedding parameters by a factor of 30. In conclusion, our method has stable compression ability and can maintain or slightly improve the model performance under different word embedding sizes.

Similar to other compression methods, our method introduces some computational overhead when generating word embeddings. However, this limitation is light according to the experimental analysis in Appendix G.

## 6   Conclusion

We propose MorphTE which combines the prior knowledge of morphology and the compression ability of tensor products to learn high-quality and space-efficient word embeddings. We validated our method on tasks such as machine translation, text summarization, and retrieval-based question answering. Experimental results show that MorphTE can handle different languages and tasks well. It outperforms related decomposition-based word embedding compression methods such as Word2ket [29] and TT embeddings [15], and can achieve dozens of times compression of word embedding parameters without hurting the performance of the original model.

## Acknowledgements

This work is supported in part by Natural Science Foundation of China (grant No.62276188 and No.61876129), the Beijing Academy of Artificial Intelligence(BAAI), TJU-Wenge joint laboratory funding, and MindSpore [5].

This work is also supported by Chinese Key-Area Research and Development Program of Guangdong Province (2020B0101350001) and the Guangdong Provincial Key Laboratory of Big Data Computing, The Chinese University of Hong Kong, Shenzhen.

---

[5] https://www.mindspore.cn/

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
