# OpenReview forum: "MorphTE: Injecting Morphology in Tensorized Embeddings"
_NeurIPS.cc/2022/Conference — NeurIPS 2022 Accept_

### Official Review · Reviewer_YXdx · 2022-07-11

**Rating:** 7
**Confidence:** 3
**Soundness:** 4 excellent
**Presentation:** 3 good
**Contribution:** 3 good

**Summary:**

This paper proposes to incorporate morphemes while compressing word embeddings and achieve lossless compression while decreasing the number of parameters. The authors compare their method to several compression techniques and show that even with a 40x compression ratio, the proposed embeddings are on par with the baseline, and on 20x they can outperform previous baselines.

**Questions:**

- I did not quite understand how do you train morpheme embeddings? Is it a linear layer in the transformer? And on which step do you divide a word into morphemes?
- Do you use words as inputs and split them into morphemes on the fly, or dataset is preprocessed with morphemes? Do you have any computational overhead, especially since your sequences are longer with morphemes representations?
- line 233: <...>**bpe**<...> - when and for what do you use bpe? Is this only for **Original**?
- do you still use positional embeddings? Does it somehow change compared to the original?


**Limitations:**

Limitations are discussed in analysis

**Strengths And Weaknesses:**

Strengths:
- Well-written paper
- Provide results on several tasks and multiple dataset for the task
- Embeddings can be use on resource-limited devices while still keep the performance

Weaknesses:
- Usage on resource-limited devices is used as motivation, however there is no further discussion and speed/memory comparison
- Ambiguity in terms which sometimes affects clarity: word/morpheme/bpe embeddings

---

> ### Author Response · Authors · 2022-08-02
> **Part 1: Thank you for your thoughtful feedback on our paper. We will answer your questions in detail. Our response is divided into two parts. Here is the first part.**
>
> **Q1: Discussion about speed/memory comparison.**
>
> Considering your valuable comments, we add the speed and memory analysis of our method and other word embedding compression methods.
>
> **Table 1.** Experimental results of the **time cost** for different embedding methods. *#Emb* represents the time cost of the word embedding layer, and *#Stuc* represents the time cost of other parts of the model except the word embedding layer. *P* represents the proportion of the time cost of the word embedding layer to the total time cost of the model.
> |      Method     |          |   GPU   |       |          |   CPU   |        |
> |:---------------|:--------:|:-------:|:-----:|:--------:|:-------:|:------:|
> |                 | *#Stuc/ms* | *#Emb/ms* | *`P/%`* | *#Stuc/ms* | *#Emb/ms* |  *`P/%`* |
> |     Original    |   10.6   |   0.1   | `0.9` |   92.0   |   0.1   |  `0.1` |
> | Matrix Factor.  |   10.6   |   0.2   | `1.9` |   92.0   |   1.0   |  `1.1` |
> |   Tensor Train  |   10.6   |   0.5   | `4.5` |   92.0   |   45.0  | `33.0` |
> |    Word2ketXS   |   10.6   |   0.6   | `5.4` |   92.0   |   49.0  | `35.0` |
> |     Word2ket    |   10.6   |   0.5   | `4.5` |   92.0   |   2.0   |  `2.1` |
> |     MorphTE     |   10.6   |   0.6   | `5.4` |   92.0   |   3.0   |  `3.2` |
>
> Theoretically, the original word embedding method can directly index the word vector from the word embedding matrix, and this process has almost no computational overhead. Almost all word embedding compression methods, including our method, require certain pre-operations to generate word embeddings, which actually introduces a certain computational overhead. This may slow down the model or increase time cost, but it is tolerable in most cases according to our experimental analysis below.
>
> We experimentally test the time cost of different word embedding methods on Intel E5-2698 CPU and Tesla V100 GPU. We choose the WikiQA task, repeat the test 100 times on an input sample, and average the time.
>
> As shown in Table 1, although word embedding compression methods increase the time cost of word embeddings, most of them only account for a small fraction of the total time cost. The time cost of our MorphTE accounts for 5.4\% and 3.2\% of the model on GPU and CPU, respectively. Compared with the obvious parameter reduction, this slight computational overhead is tolerable. In addition, we also found that the time cost of Word2ketxs and Tensor Train methods increased significantly on CPU devices. This may be related to the fact that these two methods have more computational operations under the condition of the same number of parameters.
>
>
> **Table 2.**  Experimental results of the **memory cost** for different embedding methods. *#Emb* represents the memory/parameter cost of the word embedding layer, and  *#Stuc* represents the memory/parameter cost of other parts of the model except the word embedding layer.  *P* represents the proportion of the memory/parameter cost of the word embedding layer to the total cost of the model.
> |      Method     |           | Parameter |       |              |    Memory   |       |
> |:---------------|:---------:|:---------:|:-----:|:------------:|:-----------:|:-----:|
> |                 | *#Stuc/M* | *#Emb/M* | *`P/%`* | *#Stuc/MiB* | *#Emb/MiB* | *`P/%`* |
> |     Original    |   3.8   |   6.3    |  `62` |     14.6     |     24.1    |  `62` |
> | Matrix Factor.  |    3.8    |    0.08   |  `2`  |     14.6     |     0.3     |   `2`   |
> |   Tensor Train  |    3.8    |    0.08   |  `2`  |     14.6     |     0.3     |   `2`   |
> |    Word2ketXS   |    3.8    |    0.08   |  `2`  |     14.6     |     0.3     |   `2`   |
> |     Word2ket    |    3.8    |    0.3   |  `7` |     14.6     |     1.1     |   `7`   |
> |     MorphTE     |    3.8    |    0.08   |  `2`  |     14.6     |     0.3     |   `2`   |
>
> We also further experimentally test the memory cost of different word embedding methods. As shown in Table 2, the memory reduction of the word embedding compression method is consistent with the parameter reduction. MorphTE can reduce the proportion of the original word embedding parameters/memory to the total model parameters/memory from 62\% to 2\%.
>
> We have also added the above experiments and analysis in Appendix C of the revised paper.
>
>
> **Q2: Do you have any computational overhead?**
>
> Referring to the answer of Q1, our method utilizes the tensor product to generate word embeddings, which does introduce a certain computational overhead. Other word embedding methods (such as Word2ket, TT embedding) also require specific computations to generate word embeddings. Notably, the computational cost of generating word embeddings is small (accounting for 3.2\% or 5.4\%) compared to the total computational cost of task-specific models, according to Table 1.

---

> > ### Author Response · Authors · 2022-08-02
> > **Part 2: Our response is divided into two parts. Here is the second part.**
> >
> > **Q3: How to train morpheme embeddings?**
> > What our method needs to train is the morpheme embedding matrix shown in Figure 3 of our paper. It is similar to the look-up table in word embedding, but it has a smaller shape. Given a word, the method first indexes the corresponding morpheme vectors from the morpheme embedding matrix according to the morpheme composition of the word, and then uses the indexed morpheme vectors to generate the word embedding through the tensor product. Next, the generated word embedding is fed into a task-specific model. Ultimately, the model will optimize the used morpheme vectors during backpropagation.
> >
> > **Q4: On which step do you divide a word into morphemes?  / Do you use words as inputs and split them into morphemes on the fly, or dataset is preprocessed with morphemes?**
> > To improve efficiency, we perform morpheme segmentation in the data preprocessing stage instead of real-time morpheme segmentation at the input layer. Specifically, we first count the different words in the corpus to get a vocabulary. Then, for each word in the vocabulary, we use the polyglot tool to segment it into morphemes, and store a morpheme index vector for the word, which indicates the positions of the word's morphemes in the morpheme embedding matrix. The morpheme index vectors of all words in the vocabulary form a morpheme index matrix shown in Figure 3 of our paper. Through the above preprocessing, at the input layer, the model does not need to carry out morpheme segmentation, just first find the index vector of the word from the morpheme index matrix, and then index the corresponding morpheme vectors through the index vector.
> >
> > **Q5: When and for what do you use bpe? Is this only for Original?**
> > (**i**) Almost all mainstream machine translation models use BPE to process parallel corpora to reduce the size of the vocabulary. The processing of BPE directly can affect the effect of translation. Therefore, we use the same BPE setting for all word embedding methods (not only Original) on the translation task, for a fair comparison.
> > (**ii**) For all methods, BPE is used in the early stage of data preprocessing. BPE usually keeps the common words and segments the rare words, and the word processed by BPE is called a bpe unit (if it is not segmented, it is a word, otherwise it is a subword.). With BPE preprocessing, the task of the translation model is no longer to translate the source word sequence into the target word sequence, but to translate the source bpe unit sequence into the target bpe unit sequence. At the same time, the vocabulary of the model is no longer composed of words, but bpe units (although most bpe units are words.). Therefore, all word embedding methods learn embedding representations for BPE units instead of words.
> > (**iii**) When dealing with the translation task of bpe unit sequence, our method is to perform morpheme segmentation on the bpe unit and generate the embedding for the bpe unit. In fact, when the bpe unit segmented by morphology is not a complete word, some morphemes generated here can be shared among different BPE units, but they may have no specific meaning. We admit that our method ignores this potential impact.
> >
> > **Q6: Do you still use positional embeddings?**
> > Positional embeddings are still used. This is the same for our method and other embedding compression methods (e.g., Word2ket). They all require specific operations to generate word vectors and input the generated word vectors to task-specific models (e.g., Transformer). Since the inputs are still word vectors, the model will still use the positional encoding if it was used before.

---

> > > ### Comment · Reviewer_YXdx · 2022-08-07
> > > **Reviewer comment after author response**
> > >
> > > Dear authors,
> > >
> > > Thank you for clarification and providing additional experiments results. I remain positive on my assessment of the paper.
> > >
> > > Best,
> > > Reviewer YXdx

---

### Official Review · Reviewer_mf8C · 2022-07-11

**Rating:** 8
**Confidence:** 3
**Soundness:** 3 good
**Presentation:** 4 excellent
**Contribution:** 4 excellent

**Summary:**

This paper introduces a new word embedding compression method with morphological augmentation, which is called "morphologically-enhanced tensorized embeddings (MorphTE)". The general idea is that, similar to how each word's meaning is a combination of its morphemes, each word embedding is a the product of the word's individual morpheme embeddings.
The authors conduct experiments on machine translation, question answering, and natural language inference tasks and show that MorphTE can compress word embedding parameters by a lot without performance loss, significantly outperforming related embedding compression methods.

**Questions:**

- What are the limitations of your work?
- How does the performance of your proposed approach depend on the performance of the morphological segmenter?

**Limitations:**

I don't see any direct negative societal impact of this work. The authors don't address limitations explicitly.

**Strengths And Weaknesses:**

Strengths:
- The method is well motivated *and* performs well.
- The paper is very clearly written.
- The experimental setup is well done and the proposed approach is compared to relevant baselines.

Weaknesses:
- No big ones; one could consider using newer NLI datasets like MNLI, but this isn't really a weakness.

---

> ### Author Response · Authors · 2022-08-02
> **Thank you for your thoughtful feedback on our paper. We will answer each of your questions in detail below.**
>
> **Q1: What are the limitations of your work?**
>
> Similar to other word embedding compression methods, the limitation of our method is to introduce some additional computational overhead. The overhead arises from the process of generating word embeddings. However, the overhead is light compared to the total computational cost of the model. We have added the relevant analysis in the Appendix C.2 of the revised paper to give readers a more comprehensive understanding of our work.
>
> **Q2: How does the performance of your proposed approach depend on the performance of the morphological segmenter?**
>
> **Table 1.** Experimental results of different morpheme segmenters on translation tasks. Original dictates the original word embeddings without compression. MorphTE w/ morfessor dictates the MorphTE we use in the paper. MorphTE w/ randomSeg indicates the MorphTE using randomSeg for morpheme segmentation.
> |         Method        | De-En | En-It | En-Es | En-Ru |
> |:---------------------|:-----:|:-----:|:-----:|:-----:|
> |        Original       |  34.5 |  32.9 |  39.1 |  31.6 |
> | MorphTE w/ morfessor |  34.9 |  32.9 |  39.1 |  31.9 |
> | MorphTE w/ randomSeg |  33.8 |  31.7 |  38.6 |  30.4 |
>
> Indeed, the performance of the morphological segmenter affects the performance of our method. High-quality morpheme segmentation could improve the performance of the method, while unreasonable segmentation could be counterproductive. To illustrate this, we add a comparative experiment on morpheme segmentation methods. Instead of using a morpheme segmenter, we design a random segmentation method called randomSeg. The randomseg does not segment words with a length of no more than three. For words with more than three characters, randomSeg randomly selects two gaps from the inside of the word to divide the word into 3 parts. Obviously, randomSeg makes little use of morpheme knowledge.
>
> As shown in Table 1, the performance of MorphTE w/  randomSeg is inferior to the original word embedding method and MorphTE w/ morfessor on four translation datasets. This shows that morpheme segmentation can affect the performance of MorphTE, and improving morpheme segmentation may further enhance MorphTE. We have added related experiments and analyses in Appendix B of the revised paper.

---

> > ### Comment · Reviewer_mf8C · 2022-08-09
> > **Thanks for your response!**
> >
> > Thanks for your response! The additional analysis using random segmentation is interesting and does show that having some sort of informed segmentation is useful. However, it would be nice to also have a slightly stronger baseline; how about BPE?

---

> > > ### Author Response · Authors · 2022-08-09
> > > **Thank you for your suggestion! We are trying our best to supplement this experiment.**
> > >
> > > Thank you for your suggestion! We are trying our best to supplement this experiment. If it can be completed before the end of the author-reviewer discussion stage (about 16 hours left), we will update the results.

---

> > > ### Author Response · Authors · 2022-08-09
> > > **We sincerely thank you for providing more feedback.**
> > >
> > > We sincerely thank you for providing more feedback. According to your suggestion, we further replaced random segmentation with BPE. By adjusting the number of iterations of BPE, the number of subwords generated by BPE is approximately the same as the number of morphemes generated by morfessor. Due to limited time, we only completed the experiment on one translation dataset.
> > >
> > > **Table 2.** Experimental results of different morpheme segmenters on De-En of translation tasks. Original dictates the original word embeddings without compression. MorphTE w/ morfessor dictates the MorphTE we use in the paper. MorphTE w/ randomSeg indicates the MorphTE using randomSeg for morpheme segmentation. MorphTE w/ BPE indicates the MorphTE using BPE for morpheme segmentation.
> > > | Method               | De-En |
> > > |----------------------|-------|
> > > | Original             | 34.5  |
> > > | MorphTE w/ morfessor | 34.9  |
> > > | MorphTE w/ randomSeg | 33.8  |
> > > | MorphTE w/ BPE       | 34.6  |
> > >
> > > As shown in Table 2, MorphTE w/ BPE achieves significantly better results than MorphTE w/ randomSeg, although it is slightly inferior to MorphTE w/ morfessor. This further show that having some sort of informed segmentation (e.g., BPE subwords or morphemes) is useful. In the future, we will further explore more effective segmentation methods to improve MorphTE.

---

### Official Review · Reviewer_uxZV · 2022-07-11

**Rating:** 5
**Confidence:** 3
**Soundness:** 3 good
**Presentation:** 3 good
**Contribution:** 2 fair

**Summary:**

This paper presents a model to determine word embeddings by composing morpheme embeddings. This is important research that mixes pre-existing knowledge and corpus-derived knowledge.


**Questions:**

How does the method deal with tensors of increasing dimensions?


**Ethics Review Area:**

["I don’t know"]

**Limitations:**


The paper fails to be clear with respect to the use of the tensor product. Indeed, the tensor product increases the dimensions of the tensors at each step. The dimension of the product of two tensors is the sum of dimensions of the two tensors, that is, if two matrices are the arguments of the tensor product, the results is a four-dimensional tensor. Hence, it is unclear how this model generalizes.

The authors should reduce the use of the bold and the italics.

**Strengths And Weaknesses:**

Pro:

- the idea is sound
- results are convincing

Cons:

- it is unclear the use of the tensor product

---

> ### Author Response · Authors · 2022-08-02
> **Part 1: Thank you for your thoughtful feedback on our paper. We will answer your questions in detail. Our response is divided into two parts. Here is the first part.**
>
> **Q1. How does the method deal with tensors of increasing dimensions? / The paper fails to be clear with respect to the use of the tensor product.**
>
> `Section 1 of the answer to Q1.`
>
> Thanks for your comment. We recognize that the previous version  indeed has caused some confusion about tensor dimensions when discussing the tensor product.
>
> Typically, there exist two types of definitions for the tensor product. Here, we give an example for the tensor product between two vectors (one-dimensional tensors), but it also holds for the tensor product between matrices (two-dimensional tensors).
>
> **Definition 1**: a tensor product between a m-sized vector and a n-sized vector results in a $m \times n$ matrix.
> $$ \mathbf{a} \otimes^1 \mathbf{b}
> =\left[\begin{array}{ll}
> a_{1}  \\\\
> a_{2}
> \end{array}\right] \otimes\left[\begin{array}{lll}
> b_{1} & b_{2} & b_{3}
> \end{array}\right]
> =\left[\begin{array}{lll}
> a_{1} b_1 & a_1 b_2 & a_1 b_3 \\\\
> a_{2} b_1 & a_2 b_2 & a_2 b_3
> \end{array}\right]
> $$
>
> **Definition 2**: a tensor product between a m-sized vector and a n-sized vector results in a $mn$-sized vector.
> $$ \mathbf{a} \otimes^2 \mathbf{b}  \\\\
> =\left[\begin{array}{ll}a_1 & a_2 \end{array}\right] \otimes \left[\begin{array}{lll}b_1 & b_2 & b_3 \end{array} \right]  \\\\
> = \left[\begin{array}{ll}
> a_1 \left[\begin{array}{lll}b_1 & b_2 & b_3 \end{array} \right] & a_2 \left[\begin{array}{lll}b_1 & b_2 & b_3 \end{array} \right]
> \end{array} \right] \\\\
> = \left[\begin{array}{llllll}a_1b_1 & a_1b_2 & a_1b_3 & a_2b_1 & a_2b_2 & a_2b_3 \end{array}\right]$$
> For Definition 2, there is also an example for the tensor product between matrices on the [wikipedia page](https://en.wikipedia.org/wiki/Tensor_product?#Tensor_product_of_linear_maps).
>
> Note that Definition 1 `increases the dimensions of the tensors at each step`  while  Definition 2  does not. As formulated in Eq. 2 in our paper, we use Definition 2 without dimension increase, as Word2ket [1] did. The difference between Definition 1 and Definition 2 is that Definition 2 further **reshapes** the RHS (right hand side) of Definition 1 (previously a two-dimensional tensor, i.e., a matrix) into a flattened vector.
>
> For example,  Eq. 4 in the paper represents a word by the tensor product of many small-shape vectors: $ v=\sum_{k=1}^{r} \bigotimes_{j=1}^{n} v_{j k} $ where $v_{jk} \in \mathbb{R}^{q}$. By using a typical tensor product definition (Definition 1), $v$ in the LHS (left hand side) should be a n-dimensional tensor with a shape of $  \underbrace{ q \times q \times \cdots,  \times q}_{n \ \textrm{times}} $. Since a word embedding is a vector in neural networks, we make use of Definition 2 so that any resulted embeddings for words are necessarily vectors. The resulted $ q^n $-sized vector by Definition 2 can be seen as a flattened form of the n-dimensional tensor by Definition 1. We have clarified it in the revised paper (see lines 126-128).
>
> Furthermore, we would like to give a concrete example, to build a word vector with a set of vectors using the tensor product. Suppose a word is composed of three morphemes (a,b,c). Its embedding is defined as a tensor product between these three  morpheme vectors as follow:
> $$\mathbf{a} \otimes \mathbf{b} \otimes \mathbf{c} \\
> =\left[\begin{array}{ll}a_1 & a_2 \end{array}\right] \otimes \left[\begin{array}{lll}b_1 & b_2 \end{array} \right] \otimes \left[\begin{array}{lll}c_1 & c_2 \end{array} \right] \\
> = \left[\begin{array}{llllll}a_1b_1 & a_1b_2 & a_2b_1 & a_2b_2 \end{array}\right] \otimes \left[\begin{array}{lll}c_1 & c_2 \end{array} \right] \\
> = \left[\begin{array}{llllllll}a_1b_1c_1 & a_1b_1c_2 & a_1b_2c_1 & a_1b_2c_2 & a_2b_1c_1 & a_2b_1c_2 & a_2b_2c_1 & a_2b_2c_2 \end{array}\right]$$
>
> We also put the above explanation  in Appendix A of the revised paper.

---

> > ### Author Response · Authors · 2022-08-02
> > **Part 2: Thank you for your thoughtful feedback on our paper.  Our response is divided into two parts. Here is the second part.**
> >
> > **Q1. How does the method deal with tensors of increasing dimensions? / The paper fails to be clear with respect to the use of the tensor product.**
> >
> > `Section 2 of the answer to Q1.`
> >
> > We state the rationale for using tensor products here.
> >
> > As formulated in Eq. 4 in our paper, using  tensor product, a $ q^n $-sized vector can be generated from $n$ vectors of size $q$. From a reductionism point of view,  a large vector is formed by small vectors through tensor product. Considering the vector dimensionality as the number of parameters, the tensor product can reduce parameter number exponentially (from $q^n$ to $n\times q$ in the above example), resulting in a large potential for compression. This is also the basis for many research works related to compression (e.g., Word2ket [1] and KroneckerBERT [2]).
> >
> > In our method, a word vector is formed by small morpheme vectors, via tensor product. Both tensor product and morpheme composition are non-commutative (e.g., $c \otimes e \neq e \otimes c$ and likewise "houseboat" $\neq$ "boathouse"). Therefore, the tensor product can naturally model the order of morphemes to distinguish "houseboat" from "boathouse".
> >
> > One might concatenate/sum/multiply morpheme vectors to build word vectors. However these approaches have  limited compression ratios, it is difficult to achieve compression beyond an order of magnitude. Moreover,  concatenation lacks effective interaction between morpheme vectors. Element-wise addition and multiplication are commutative, which is inconsistent with the non-commutative nature of the morpheme composition.
> >
> > To sum up, we believe that the tensor product is a reasonable way to model and compress word embeddings, at least in the scenario of processing the morpheme composition.
> >
> > [1] Aliakbar Panahi, Seyran Saeedi, and Tomasz Arodz. word2ket: Space-efficient word embeddings inspired by quantum entanglement. In ICLR, 2020.
> > [2] Marzieh S. Tahaei and Ella Charlaix et al. Significant Compression of Pre-trained Language Models Through Kronecker Decomposition and Knowledge Distillation. In NAACL-HLT, 2022.
> >
> >
> > **Q2. The authors should reduce the use of the bold and the italics.**
> > After careful consideration of your suggestions, we have revised the paper and reduced the use of bold and italics, especially on pages 2 and 7.

---

### Author Response · Authors · 2022-08-08
**We sincerely thank you for providing comments and reviews, and look forward to more feedback.**

Dear reviewers,

We sincerely thank you for contributing time to provide comments and reviews. Especially thank reviewer YXdx for commenting on our response and remaining positive about this paper.

There are less than 2 days left for the stage of the author-reviewer discussion. We hope to further discuss with you whether or not your concerns have been addressed.

Based on the first-round reviews, we have revised our paper in the revision to respond to your concerns and comments.

(1) We have clarified the use of tensor products and added specific examples in the appendix of the revised paper for readers to better understand.

(2) We have added the limitations of our work and analyzed the impact of morphological segmenters on our method in the revised paper.

(3) We have added the speed and memory analysis of our method and other word embedding compression methods in the revised paper. The results show that our method can significantly reduce the parameters and memory usage with a tolerable time cost, while achieving better effectiveness than other compression methods.

(4) We have clarified the details of morpheme segmentation, morpheme embedding training, etc.

Best,
Authors of Paper 11117

---

### Meta-Review · Area_Chair_yvyh · 2022-08-25

**Recommendation:** Accept
**Confidence:** Certain

**Metareview:**

The paper presents a method to compress word embeddings using morphologically-enhanced tensorized embeddings. The main idea is decomposing a word embedding into a tensor product of multiple small vectors. This idea has been explored in previous work [1], the main contribution of this is using *morphological segmentation*, aiming to capture morphological features.

The proposed method is compared to prior work on multiple tasks (machine translation, qeustion answering, etc), overall exhibiting strong performance in terms of compression rate & performance (though sometimes gains are pretty smaller compared to Word2ket (e.g., Table 2)).

During the discussion phase, the authors provided the results with random, non-morpholgically inspired segmenation, which was useful. I’d recommend putting them into the main paper. Experimenting on morphologically rich languages (e.g., Finnish) would be useful. As well as sensitive study towards hyper parameters (e.g., dimensionality of morpheme embeddings) which can easily impact the compression rate & performance.

Overall, the reviewers and AC is positive about the paper.

[1] word2ket: Space-efficient Word Embeddings inspired by Quantum Entanglement
https://arxiv.org/abs/1911.04975

**Award:**

No

---

### Decision · Program_Chairs · 2022-09-14

Accept